# A matter of size: Comparing IV and OLS estimates

**Riccardo Ciacci** *

Department of Economics, Universidad Pontificia Comillas, Spain, Madrid

* riccardo.ciacci@eui.eu

## Abstract

Sizeable differences between OLS (Ordinary Least Squares) and IV (Instrumental Variables) estimates might be interpreted in the literature as evidence that the instrument is not valid. Yet, to the best of our knowledge, this comparison is carried out using only the OLS coefficient as a benchmark and does not account for statistical measures or information from the OLS regression. This paper establishes a framework where [1]'s methodology might be used to compare objectively OLS and IV estimates. This methodology offers evidence to support or discard IV estimates with respect to the OLS regression.

**Data availability statement:** All data are in the manuscript and/or Supporting information files.

**Funding:** The author(s) received no specific funding for this work.

**Competing interests:** The authors have declared that no competing interests exist.

## Introduction

Instrumental variables techniques are a fundamental method in the econometrics toolkit in order to solve both issues of endogeneity and measurement error [2,3]. A common approach in econometrics is to interpret substantial differences between the magnitudes of Ordinary Least Squares (OLS) and Instrumental Variables (IV) coefficients as evidence against the validity of the instrument. The intuitive appeal of this strategy is that the OLS regression can be informative about the true effect the researcher wants to estimate. Nonetheless, in the literature, to the best of our knowledge, there is no formal methodology to compare these two estimates. This paper implements [1] methodology to make such a comparison. This methodology is related to [4] in its usage of partial least squares and to [5] in its usage of reliance on selection on observables to recover selection on unobservables. IV is an econometric methodology primarily used to estimate a causal effect when the explanatory variable is considered endogenous or in cases of measurement error of that variable. This paper addresses scholars using IV in the first case mentioned above.

[1] makes use of information from the OLS regression - such as, inclusion of controls, size of variances and movement of $R^2$, etc.- to estimate a set of values where the true treatment effect should lie. The size of such set depends on how "informative" the observables are about the unobservables according to the researcher. Consequently, this methodology allows the researcher to compute a parameter to develop a formal bounding argument. This parameter is known as *coefficient of proportionality* and measures the relative size of the proportionality between selection on observables and unobservables.

This article argues that this methodology might be useful to make objective comparisons between IV and OLS estimates. To this end, this paper suggests that larger (smaller) values of the coefficient of proportionality are evidence against (in favour of) IV estimates.

The main contribution of this paper is to suggest an implementation of the methodology developed in [1] through which scholars can compare IV and OLS estimates. First, this paper is related to several recent papers comparing the relative size of such two estimates without a formal methodology (see, among others, [6–9]). Second, this paper is also related to a branch of the literature that uses observables to asses the bias generated by unobservables in OLS settings [1,4,5,10–13].

The rest of the paper is divided in the following sections. First, we introduce the methodological framework intuitively and suggests how to implement it. Then, we apply the methodology to different settings. Finally, we offer concluding remarks. S1 Appendix Sects A and B briefly explains how to use this methodology in Stata.

## Framework

Let the population regression function (hereinafter, PRF) be given by the panel data regression:

$$y_{ih} = \alpha_1 + \beta_1 d_{ih} + \gamma w_{ih} + \theta_1 X_{ih} + \varepsilon_{1ih} \tag{1}$$

where, for each variable, $ih$ denotes unit $i$ at time $h$. $d$ is the (scalar) treatment we are interested in, $w$ is the vector of unobserved controls and $X$ is the vector of observed controls. $\varepsilon$ is the error term. Note that this framework allows to consider only an unobserved control $w$. Computations will follow depending on the considered unobserved control. This is a limitation of this study. Given the nature of $w$, we can only estimate

$$y_{ih} = \alpha_2 + \beta_2 d_{ih} + \theta_2 X_{ih} + \varepsilon_{2ih} \tag{2}$$

Note that subscript 1 denotes terms in Eq (1), while subscript 2 denotes terms in Eq (2). If we are in a setting where the assumptions exposed in [1] are plausible we can take into consideration the proportional selection relationship given by:

$$\delta \frac{Cov(d_{ih}, X_{ih})}{Var(X_{ih})} = \frac{Cov(d_{ih}, w_{ih})}{Var(w_{ih})} \tag{3}$$

which holds for some $\delta \neq 0$.

Now consider the simple univariate setting. We know that in the case where $w$ is the only control, and it is omitted from the regression, the omitted variable bias is given by:

$$\hat{\beta}_2 = \beta_1 + \gamma \frac{Cov(d_{ih}, w_{ih})}{Var(d_{ih})} \tag{4}$$

Recall that if the instrument is valid the IV coefficient consistently estimates $\beta_1$. Therefore, given Eq (3), we can compute how large $\delta$ needs to be to support the difference in size between $\beta_2$, the OLS estimates, and the candidate to be the true effect $\beta_1$, the IV estimates. Plugging Eq (3) into Eq (4) and solving for $\delta$ we can easily get:

$$\delta = (\hat{\beta}_2 - \beta_1) \frac{Var(d_{ih})Var(X_{ih})}{\gamma Var(w_{ih})Cov(d_{ih}, X_{ih})} \tag{5}$$

The computations above are an approximation to develop intuition. S1 Appendix Sect C computes both the univariate and multivariate cases. These results help clarify which statistical objects determine the sign of $\delta$. Moreover, Table 1 summarizes the variables involved, and their corresponding assumptions, in the computations. A similar table for the multivariate case is shown in the S1 Appendix, labelled as Table A.1.

A low coefficient of proportionality supports the size of the IV estimates. Put it differently, a low $\delta$ means that not much selection on unobservables is needed to support that the true effect is the one estimated by the IV regression. Intuitively, $\delta$ tells us how large selection on unobservables, compared to observables, needs to be to support that the "true effect" has the size of the IV estimates. Since this analysis takes into account inclusion of controls, size of variances and movement of $R^2$, among other things, large values of $\delta$ might imply either that the instrument is not valid or that there are heterogeneous effects and the IV estimates are estimating them for a subpopulation [14]. In this circumstance (i.e. a large coefficient of proportionality) to distinguish between the two cases this methodology might be complemented using [15] methodology and carrying out an analysis, as in [8], to explore whether in the data there is empirical evidence supporting the IV is estimating heterogeneous effects for a subpopulation. While a large proportionality coefficient may raise concerns about instrument validity, this methodology alone cannot confirm it. Complementary analyses are required to properly assess validity, as the current approach is not designed for this purpose.

At this stage, it is pivotal to develop intuition about the plausibility of Eq (3) depending on the setting. If the researcher believes that the considered covariates mantain a proportional selection relationship with the unobserved variables, then Eq (3) is plausible and estimating $\delta$ is informative about the relative size of the IV estimates with respect to the OLS ones. On the contrary, unobservables that do not follow such a proportional selection relationship cannot be addressed by this methodology.

In empirical settings in order to compute the identified sets it might be important to establish the sign of $\delta$. In the simplified equation presented above (i.e. Eq (5)) the sign of $\delta$ only depends on the sign we assume $\gamma$ has, since variances are positive and we can estimate from the data all the other objects of the equation. This formula is a simplification drawn from the univariate setting, still the intuition is clear: giving the known sign of $(\hat{\beta}_2 - \beta_1)$ and $Cov(d_{ih}, X_{ih})$, we can guess the sign of $\delta$ depending on the sign of the effect we think the omitted variable has in the main regression. Once we have the sign we can compute the identified set for coefficients of proportionality with that sign to check how the bounds of the set vary as the coefficient of proportionality varies.

**Table 1. Summary: Univariate case.**

| Symbol | Meaning | Role / Assumption |
|---|---|---|
| $y_{ih}$ | Outcome variable | Dependent variable |
| $d_{ih}$ | Treatment variable | Scalar variable of interest |
| $X_{ih}$ | Observed control | Single observed covariate |
| $w_{ih}$ | Unobserved control | Single unobserved covariate |
| $\beta_1$ | True treatment effect | Causal effect in full model |
| $\beta_2$ | OLS estimate | Biased estimate from partial model |
| $\gamma$ | Effect of unobservable | Coefficient of $w_{ih}$ in full model |
| $\delta$ | Coefficient of proportionality | Measures relative selection on |
|  |  | observables vs. unobservables |
| $\varepsilon_{1ih}, \varepsilon_{2ih}$ | Error terms | i.i.d. with mean zero |

Lastly, a key input to estimate the identified set is the selection of the value that the $R^2$, statistic would take in a hypothetical full regression, a regression with both observables and unobservables (i.e. Eq (1)). [1] denotes such value as $R_{max}$. To select $R_{max}$ prior knowledge of the setting is crucial. [1] discusses this issue in detail. Namely, whether the researcher believes the full regression can explain the outcome variable completely. If this is the case $R_{max}$ is set to 1.

**Implementation:** This implementation is similar in spirit to the "Statements about $\delta$" subsection discussed in section 3.4 of [1]. Unlike the afore-mentioned subsection, we do not have a suggested upper bound for $\delta$. However, we would expect $|\delta| > 1$ since in our framework the IV regression is run since the OLS regression suffers omitted variable bias and cannot pin down a causal relationship. Hence, values of $|\delta|$ slightly larger than 1 are considered low values of the coefficient of proportionality.

1. Set the sign of the coefficient of proportionality $\delta$ depending on the regression and the omitted variable the researcher wants to adress. Eq (5) can be helpful to this end.
2. Set $R_{max}$ according to knowledge of the setting. If the researcher believes the full regression can explain the outcome variable completely set $R_{max} = 1$. If this is not the case, the researcher might report results for a plausible range of $R_{max}$ values (e.g., $R_{max} = \min(1.3 \times \tilde{R}^2, 1)$, or $R_{max} = 0.8, 0.9, 1$). Presenting a table or a plot of $\delta$ against $R_{max}$ for a fixed IV coefficient is highly informative and transparent.
3. Estimate the coefficient of proportionality $\delta$, with the sign computed above, that could explain the IV estimates.
4. This value measures the minimum amount of selection on unobservables (compared to observables) needed for the treatment effect to have the same size of the IV estimates (i.e. for the IV estimates to lie in the identified set). Discuss whether this value is excessively large.

## Potential limitations

Eq (3) is most likely violated in the following cases:

1. When the included observables ($X$) are poor proxies for the type of unobservables ($w$) that cause the omitted variable bias.
2. If observables and unobservables influence the treatment through different economic or social mechanisms, and the degree of selection on such mechanisms differs.
3. If the observed controls ($X$) are themselves measured with a non-random error, the estimated $\text{Cov}(d_{ih}, X_{ih})$ will be attenuated toward zero. This distortion might affect the proportional scaling on the left-hand side of the equation.

## Empirical validation

This section makes use of different data sets to explore how to use the methodology introduced in the previous section. Results show that comparing the OLS and IV estimates simply due to their relative size is not enough to assess whether the latter is a credible estimate of the true effect.

## Simulated data example

This section shows the results of simulating 10,000 samples with 10,000 observations each. The corresponding do file of these computations might be found and downloaded as extra

material. The objective of this section is threefold. First, it investigates whether IV estimates are reliable even when they are considerably different in size compared to the OLS ones. Second, it evaluates how accurately the procedure analyzed in this paper (i.e. in subsection "Implementation") computes $\delta$. Third, it explores how the coefficient intervals estimated with the methodology developed in [1] perform in this IV setting.

With this aim in mind, we make the following assumptions:

1. There is only one observable and one unobservable variable.
2. The unobservable is negatively correlated with the observable.
3. In absolute value the correlation between the unobservable and the treatment is always larger than the correlation between the observable and the treatment.
4. The PRF effect of the unobservable on the outcome is larger than the treatment effect.
5. The variance of the unobservable is larger than the one of the observable

Fig 1 shows that high values of the IV/OLS ratio do not necessarily imply the instrument is not valid. IV variables of the simulation fulfill the usual requirements (i.e. exogeneity, exclusion restriction and strong first stage). The figure plots the IV/OLS ratio on the vertical axis and the Treatment effect/OLS ratio on the horizontal axis. The red line is the function given by the equality of the two axis. The figure shows data are distributed along this line, pointing that high values of the IV/OLS ratio are "justified" by high values of the Treatment effect/OLS ratio.

Since this section deals with simulated data we both know the true delta and the estimated delta using our methodology; we respectively denote those two by $\delta$ and $\hat{\delta}$. To this extent, Tables 2 and 3 respectively show descriptive statistics of $\delta$ and $\hat{\delta}$ for different values of $|\hat{\delta}|$. As it might be expected, comparing descriptive statistics suggests that the two are more likely to be similar if $|\hat{\delta}|$ is low. Put it differently, these results suggest that $\hat{\delta}$ is more likely to be informative about $\delta$ if $|\hat{\delta}|$ takes low values. To this extent, row 1 of Table 2 shows that out of 10,000

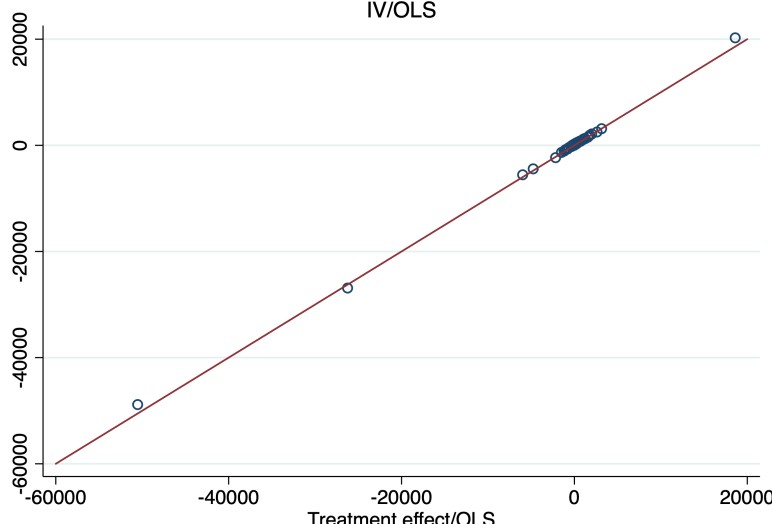

**Fig 1. Comparison ratio IV/OLS vs treatment effect/OLS.** Notes: This figure plots the ratio between the IV and OLS estimates on the vertical axis and the ratio between the Treatment effect and OLS estimates on the horizontal axis. The red line is the function given by the equality of the two axis. Data are distributed on this line for any value of the ratios suggesting there is no difference between high and low values of the IV/OLS ratio.

**Table 2. Comparison of delta: True (i.e. simulated) versus estimated.**

| Obs | Mean | Std. Dev. | Min | Max | $|\hat{\delta}| <$ |
|---|---|---|---|---|---|
| 3,593 | 1.1 | 0.49 | 0.18 | 2.45 | 1 |
| 6,753 | 1.06 | 0.49 | 0.18 | 2.45 | 2 |
| 8,198 | 1.05 | 0.49 | 0.18 | 2.45 | 3 |
| 8,945 | 1.04 | 0.49 | 0.18 | 2.45 | 4 |
| 9,295 | 1.03 | 0.49 | 0.18 | 2.45 | 5 |
| 9,762 | 1.01 | 0.49 | 0.18 | 2.45 | 10 |
| 10,000 | 1 | 0.49 | 0.18 | 2.45 | 100 |

Notes: This table presents descriptive statistics for $\delta$ across varying values of $\hat{\delta}$.

**Table 3. Comparison of delta: Estimated versus true (i.e. simulated).**

| Obs | Mean | Std. Dev. | Min | Max | $|\hat{\delta}| <$ |
|---|---|---|---|---|---|
| 3,593 | 0.82 | 0.58 | -1 | 1 | 1 |
| 6,753 | 1.37 | 0.73 | -1 | 2 | 2 |
| 8,198 | 1.57 | 0.8 | -1 | 3 | 3 |
| 8,945 | 1.73 | 0.93 | -1 | 4 | 4 |
| 9,295 | 1.83 | 1.05 | -1 | 5 | 5 |
| 9,762 | 2.06 | 1.48 | -1 | 10 | 10 |
| 10,000 | 2.27 | 1.98 | -1 | 10.9 | 100 |

Notes: This table shows descriptive statistics of $\hat{\delta}$ as it increases.

observations 3,593 have $|\hat{\delta}| < 1$, for these observations on average $\delta = 1.1$ while standard deviation, min and max are respectively 0.49; 0.18 and 2.45. Results of these last three statistics are stable as $|\hat{\delta}|$ increases.

If we restrict to observations where $|\hat{\delta}| < 2$, as row 2 shows, we are left with 6,753 cases out of 10,000 observations. In this case even if the standard deviation, min and max are stable, the average $\delta$ decreases to 1.06. Since our estimates indicate $|\hat{\delta}| < 2$ this guess is less precise than row 1, where the average is 1.1 and the guess $|\hat{\delta}| < 1$. The same pattern is true as $|\hat{\delta}|$ is bounded above by a larger number, as it is shown from row 3 onwards.

By the same token, Table 3 shows summary statistics of the $\hat{\delta}$ associated to the corresponding row of Table 2. We can observe that as the upperbound of $|\hat{\delta}|$ rises, its difference in absolute value with the average increases as well. Also this piece of evidence suggests that larger $\hat{\delta}$ are worse guesses.

There might be concerns about how likely is that the sign of the coefficient of proportionality $\delta$ is correctly estimated by $\hat{\delta}$. To this end, Table 4 shows the frequency and relative share of samples for which the sign of the coefficient of proportionality $\delta$ is correctly estimated (i.e. $sign(\hat{\delta}) = sign(\delta)$) as $\hat{\delta}$ increases. This table indicates, out of each category, for over 90% of the cases the sign of the coefficient is estimated correctly.

We might wonder how likely it is to identify a set of values such that the IV estimates lie within. To this end, Fig 2 plots the distribution of the treatment effect and the three estimators considered in this paper: OLS, IV and Oster for the samples in which the sign of the coefficient of proportionality $\delta$ is correctly estimated (i.e. $sign(\hat{\delta}) = sign(\delta)$), 9,668 samples out of 10,000.

There are at least two features to note. First, generally the IV estimates are those that most accurately estimate the treatment effects. Second, by and large OLS in this setting (i.e. endogeneity) performs poorly.

**Table 4. Frequency.**

| Frequency | Total | Percentage | $|\hat{\delta}| <$ |
|---|---|---|---|
| 3,261 | 3,593 | 0.91 | 1 |
| 6,421 | 6,753 | 0.95 | 2 |
| 7,866 | 8,198 | 0.96 | 3 |
| 8,613 | 8,945 | 0.96 | 4 |
| 8,963 | 9,295 | 0.96 | 5 |
| 9,430 | 9,762 | 0.97 | 10 |
| 9,668 | 10,000 | 0.97 | 100 |

Notes: This table reports the frequency and proportion of samples in which the sign of $\delta$ is correctly estimated (i.e. $sign(\hat{\delta}) = sign(\delta)$) as $|\hat{\delta}|$ increases.

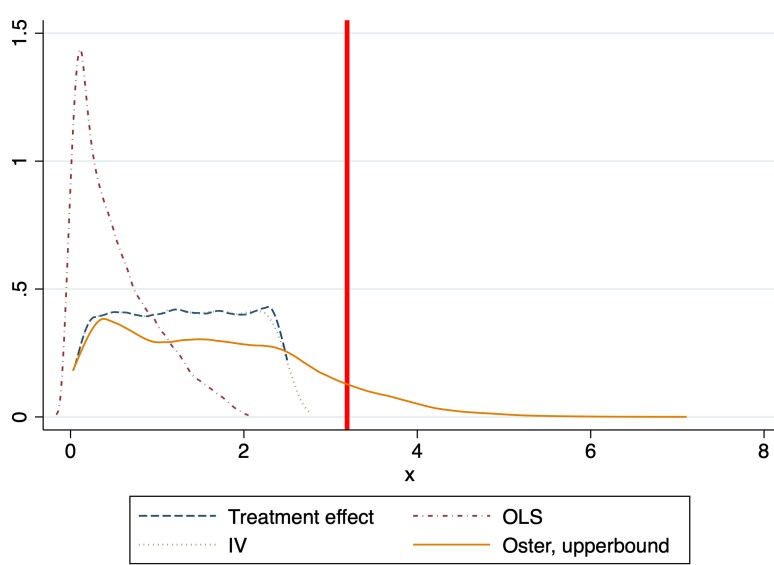

**Fig 2. Distribution of the treatment effect and estimators.** Notes: This figure plots the distribution of the treatment effect and the three estimators considered in this paper: OLS, IV and Oster for the samples in which the sign of the coefficient of proportionality $\delta$ is correctly estimated (i.e. 9,668 samples out of 10,000). Given the structure of the samples the OLS and Oster estimators are respectively a lower- and upper- bound of the Treatment effect. This figure indicates that, in this setting, the Oster estimator is more accurate than the OLS estimator. In particular, this is the case for 90% of the samples.

Given the structure of the samples the OLS and Oster estimators are respectively a lower- and upper- bound of the Treatment effect. The vertical line cuts the distribution of Oster estimators at the 90% percentile. This figure supports that in this setting the Oster estimator is a reasonable upper-bound of the Treatment effect. Indeed, the Oster estimator is closer to the Treatment effect, than the OLS estimator, for 90% of the samples considered.

**Practical guidelines for applied researchers.** Our simulation results provide concrete guidance for interpreting the coefficient of proportionality $\delta$ in empirical applications. To this extent, Fig 3 and Table 5 summarize key steps, thresholds and actions based on our findings.

The proposed thresholds for $\delta$ are guidelines, not strict rules. Their interpretation depends critically on the specific empirical context and the strength of the available observables. A large $|\hat{\delta}|$ does not conclusively prove instrument invalidity; it may also indicate the instrument identifies a heterogeneous local average treatment effect . Further analysis is essential to distinguish between these two explanations.

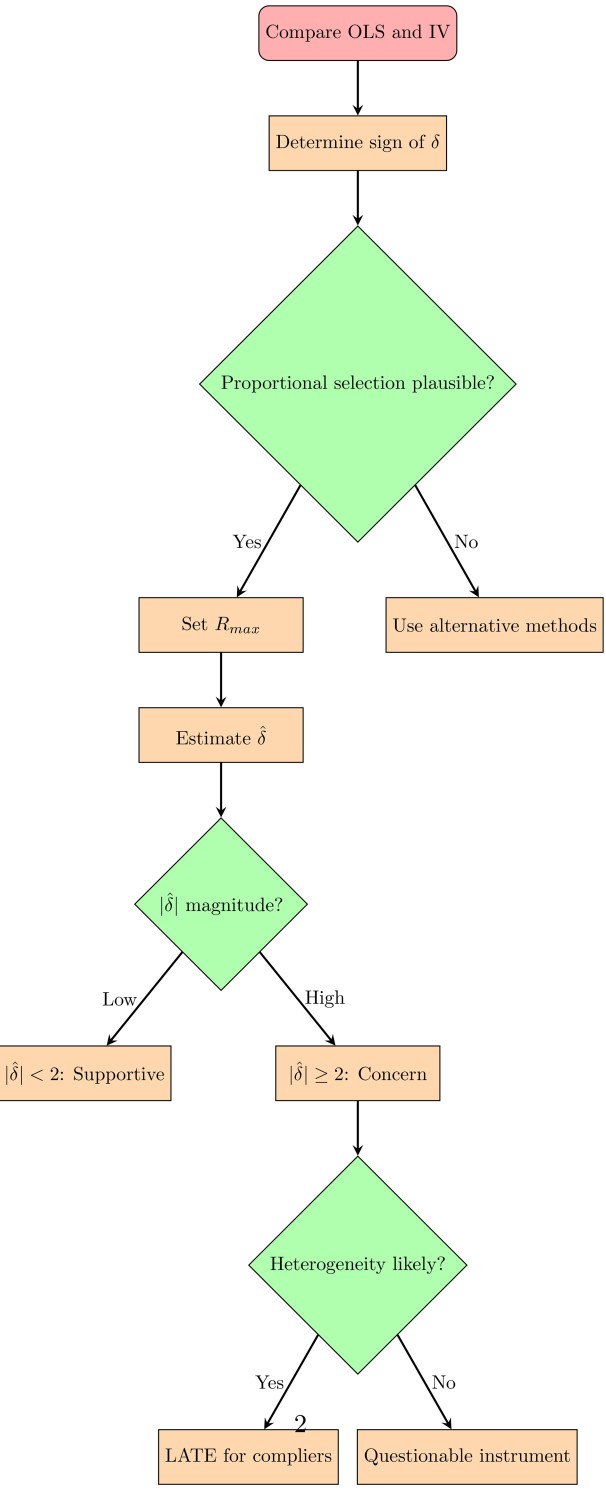

**Fig 3. Decision tree for estimating and interpreting $\delta$.**

**Table 5. Checklist for interpreting the coefficient of proportionality $\delta$.**

| Magnitude of $|\hat{\delta}|$ | Interpretation and Implication | Recommended Action |
|---|---|---|
| $|\hat{\delta}| < 1$ | Strong supportive evidence. The IV estimate is plausible even if selection on unobservables is weaker than on observables. | Proceed with cautious optimism. The instrument is likely valid regarding size. |
| $1 \leq |\hat{\delta}| < 2$ | Moderate supportive evidence. The IV estimate requires selection on unobservables to be similar to or moderately larger than on observables. | This is a common and often acceptable range. Report results but note the assumption. |
| $2 \leq |\hat{\delta}| < 3$ | Cause for concern. The IV estimate requires selection on unobservables to be more than twice as large as on observables. | Exercise significant caution. Conduct thorough robustness checks on $R_{max}$ and the proportional selection assumption. |
| $|\hat{\delta}| \geq 3$ | High cause for concern. The estimated $\hat{\delta}$ becomes a progressively worse guess of the true $\delta$ (see Table 3). The required selection on unobservables is very large. | The size of the IV estimate is likely suspect. Strongly consider alternative explanations: (1) instrument invalidity or (2) that the IV captures a highly heterogeneous Local Average Treatment Effect. |

## Observational data

**Example 1.** This section applies the presented methodology to the seminal paper [16]. The main threat to their OLS estimates is that there might be unobservables (mainly geographic and climatic features) positively correlated with economic growth and the treatment. For example consider temperate weather or natural resources. The paper also discusses a number of control variables which could affect economic growth but which might be affected by the treatment variable (such as current religion, diseases, etc.). This section is not meant to take into account these controls (*bad* controls). Given that IV estimates are greater than OLS estimates, there might be the concern that OLS regressions give rise to downward biased estimates. In this setting, a naive comparison between the two highlights that the IV estimates are at most almost three times larger than OLS estimates. This setting exemplifies a case where, even if the IV estimates are only slightly larger than the OLS estimates, a large $\delta$ is required to justify the IV estimates.

Table 6 considers the simplest IV regression with controls of [16]. In [16] the output of this regression is displayed in column (7) of Table 4. In this regression model the authors regress the log of 1995 GDP per capita on average protection against expropriation risk between 1985–1995, instrumented with settler mortality, and control for continent dummies (i.e. continent fixed effects). Table 6 has the same format and assumptions (i.e. $R_{max} = 1$) of Table 9.

Further, in the setting of [16] it makes sense to suspect that:

- Taking into account previous literature [17], unobservables (e.g. temperate weather) correlated with observables (e.g. latitude/geographic dummies) are positively correlated with the outcome variable (*y* following the nomenclature of Section Framework), 1995 GDP per capita.
- Given results from the first-stage regression (Table 3 of [16]) and previous literature [18], the treatment variable, *d* following the nomenclature of Section Framework, is positively correlated with control variables.

Hence, $\delta$ has negative sign. This is a result using: the IV and OLS estimates from [16], the two signs of the correlation discussed in this subsection and Eq (5). The last two rows of

**Table 6. Comparison of OLS and IV estimates.**

| (1) | (2) |
|---|---|
| $\delta$ | Coefficient |
| -1 | 0.62030 |
| -2 | 0.72640 |
| -3 | 0.76790 |
| -4 | 0.78880 |
| -5 | 0.80130 |
| -10 | 0.82630 |
| -20 | 0.83870 |
| -50 | 0.84620 |
| -100 | 0.84870 |
| -1,000 | 0.85100 |
| Technique | Coefficient |
| OLS | 0.42380 |
| IV | 0.98220 |

Notes: This table shows the bounds of the identified set using data from [16]. In [16] the output of this regression is displayed in column (7) of Table 4. Column (1) shows either the value of the coefficient of proportionality $\delta$ or the technique used. Column (2) shows the value of the associated estimated coefficient. For each coefficient of proportionality $\delta$ the bounds of the interval are given by the OLS estimates and the coefficient estimated with that coefficient of proportionality $\delta$.

Table 6 show the OLS and IV estimates of this regression. Table 6 shows that, in spite of the IV estimates being less than three times larger than the OLS one, there is no coefficient of proportionality $\delta$ between -1 and -1000 that could rationalize the IV estimates. Also in this case the sign of the coefficient of proportionality $\delta$ is consistent with Eq (5). Yet, similar results hold even if $\delta$ is suspected to have positive sign. Tables are available upon request.

There might be the concern that inclusion of controls could easily affect these results. To this extent, Table 7 considers the same regression model of Table 6 but where latitude (i.e. a variable taking into account the distance from the equator scaled between 0 and 1) is included. This regression corresponds to column (8) of Table 4 in [16]. As with geographic binary variables, latitude is correlated with unobservables and positively correlated with the treatment variable.

Table 7 shows that inclusion of this control decreases slightly the OLS estimates and increases slightly the IV estimates. Yet, results are unchanged. There is no coefficient of proportionality $\delta$ between -1 and -1000 that could rationalise the IV coefficient.

Lastly, Table 8 considers one of [16]'s most demanding specifications. The output of this regression is displayed in Table 6, column (9) of [16]. In this regression, the authors test the robustness of their results. Namely, they add a large set of controls to their regression model to check how their main coefficient changes. In this case the OLS and the IV estimates decrease, yet also the upper bound of the identified sets of Tables 6 and 7 goes down. All the same, results do not change: no coefficient of proportionality $\delta$ between -1 and -1000 can rationalise the IV coefficient.

This analysis highlights that the IV estimates of these regressions were too large compared to the OLS estimates. This finding casts doubt on the IV estimation. However, the interpretation of these results is not straightforward, it requires prior knowledge of the setting we are analyzing. Specifically the researcher needs to determine whether the assumptions made to use this methodology are plausible. For example, it might be that selection on observables is uninformative about selection on unobservables, yet, if this is the case there is no point in checking how the main coefficient changes after inclusion of controls. It is important to recall

**Table 7. Comparison of OLS and IV estimates.**

| (1) | (2) |
|---|---|
| $\delta$ | Coefficient |
| -1 | 0.62800 |
| -2 | 0.72210 |
| -3 | 0.75520 |
| -4 | 0.77180 |
| -5 | 0.78170 |
| -10 | 0.80150 |
| -20 | 0.81140 |
| -50 | 0.81730 |
| -100 | 0.81930 |
| -1,000 | 0.82110 |
| Technique | Coefficient |
| OLS | 0.40130 |
| IV | 1.10710 |

Notes: This table shows the bounds of the identified set using data from [16]. In [16] the output of this regression is displayed in column (8) of Table 4. Column (1) shows either the value of the coefficient of proportionality $\delta$ or the technique used. Column (2) shows the value of the associated estimated coefficient. For each coefficient of proportionality $\delta$ the bounds of the interval are given by the OLS estimates and the coefficient estimated with that coefficient of proportionality $\delta$.

**Table 8. Comparison of OLS and IV estimates.**

| (1) | (2) |
|---|---|
| $\delta$ | Coefficient |
| -1 | 0.47760 |
| -2 | 0.53820 |
| -3 | 0.57150 |
| -4 | 0.59130 |
| -5 | 0.60420 |
| -10 | 0.63200 |
| -20 | 0.64690 |
| -50 | 0.65610 |
| -100 | 0.65920 |
| -1,000 | 0.66200 |
| Technique | Coefficient |
| OLS | 0.37210 |
| IV | 0.71270 |

Notes: This table shows the bounds of the identified set using data from [16]. In [16] the output of this regression is displayed in column (9) of Table 6. Column (1) shows either the value of the coefficient of proportionality $\delta$ or the technique used. Column (2) shows the value of the associated estimated coefficient. For each coefficient of proportionality $\delta$ the bounds of the interval are given by the OLS estimates and the coefficient estimated with that coefficient of proportionality $\delta$.

that, a further explanation that the methodology developed in this paper cannot discard, is that the effects are heterogeneous for the subpopulation affected by the instrument.

**Example 2.** In this section we use data from [19] where we estimate the effect of penalizing the purchase of prostitution on rape. Given selection into treatment in this setting, reverse causality and omitted variable bias are the main concerns connected to endogeneity of the treatment variable. Reverse causality arises from the concern that past values of rape could affect fines for sex purchase: prostitutes might prefer to locate in regions with low rapes. Omitted variable bias arises since we cannot control for variables that displaces prostitutes.

Such variables are negatively correlated with fines and positively with rape, leading OLS estimates to be downward biased.

To address these issues we use two instruments that exploit variation in flights to proxy access to sex tourism. The key identification assumption is that variation in the offering of intercontinental flights is independent of rape and fines for sex purchase patterns. In other words, the choice of flight companies to offer relatively more intercontinental flights does not depend on any reason connected to rape or fines for sex purchase. This seems plausible since to the best of our knowledge there is no evidence of flight companies that choose to offer more flights due to any reason connected to crime patterns.

My structural regression is:

$$log(1 + rape_{rmy}) = \beta fines_{rmy} + \alpha_r + \alpha_m + \alpha_y + \alpha_r * y + \gamma officers_{ry} + \varepsilon_{rmy} \tag{6}$$

where $r$ stands for region, $m$ for month and $y$ for year. The dependent variable is $log(1 + rape_{rmy})$ we use the variable in logs due to the dispersion of the distribution of rapes and $log(1 + y)$ since rape may take value 0, $fines_{rmy}$ is the number of fines for sex purchase issued by police officers in region $r$ in month $m$ and year $y$; $\alpha_r$, $\alpha_m$, $\alpha_y$ are respectively fixed effects for region, month and year; $\alpha_r * y$ is a region-year trend and the control variable $officers_{ry}$ is the number of police officers in region $r$ in year $y$ since police officers are hired regionally every year.

Therefore, as for Eq (3) in this setting, $d_{ih}$ is fines for sex purchase, $X_{ih}$ is the number of police officers and $w_{ih}$ are the omitted variables and past values of the outcome that displace prostitutes. In order to use [1]'s methodology the researcher needs to assess whether Eq (3) is plausible in this setting. In this case this assumption boils down to whether we believe that selection on unobservables that displaces prostitutes is proportional to the number of officers. To this extent, where there were higher past values of rape, currently there should be fewer prostitutes but more officers. Hence, the two appear to be oppositely related (i.e. negative coefficient of proportionality $\delta$). In other words, in this setting, it is reasonable to assume:

- Taking into account previous literature (see, inter alia, [20,21]), unobservables (e.g. past values of rape) correlated with observables (e.g. officers) are positively correlated with the outcome variable ($y$ following the nomenclature of Framework), rape.
- The treatment variable: fines for sex purchase, $d$ following the nomenclature of Section Framework, is positively correlated with control variables, given results from the first-stage regression (available upon request).

Given the difference between the OLS and IV estimates, the sign of these two correlations imply the coefficient of proportionality $\delta$ is negative in this case.

Table 9 shows the coefficients estimated by [1]'s methodology setting a negative sign of the coefficient of proportionality $\delta$ and $R_{max} = 1$. The first ten rows of Table 9 show how large the coefficient of proportionality $\delta$ needs to be, column (1), to identify a coefficient of the size of column (2). In particular, for each value of $\delta$ the identified set is given by the coefficient estimated using [1]'s methodology and the OLS estimate.

The next-to-last row of Table 9 displays the OLS coefficient, whereas the last row shows the IV coefficient. Note that the latter is about 14 times larger than the former. At first sight this difference might appear considerable, however, taking into account Table 9, [1]'s methodology highlights that a negative coefficient of proportionality with size 1.16 is enough to identify

**Table 9. Comparison of OLS and IV estimates.**

| (1) | (2) |
| --- | --- |
| $\delta$ | Coefficient |
| -1 | 0.01620 |
| -2 | 0.03490 |
| -3 | 0.07020 |
| -4 | 0.11010 |
| -5 | 0.12640 |
| -10 | 0.15010 |
| -20 | 0.15960 |
| -50 | 0.16490 |
| -100 | 0.16660 |
| -1000 | 0.16810 |
| Technique | Coefficient |
| OLS | 0.00130 |
| IV | 0.01890 |

Notes: This table shows the bounds of the identified set using data from [19]. Column (1) shows either the value of the coefficient of proportionality $\delta$ or the technique used. Column (2) shows the value of the associated estimated coefficient. For each coefficient of proportionality $\delta$ the bounds of the interval are given by the OLS estimates and the coefficient estimated with that coefficient of proportionality $\delta$.

a set that includes the IV estimates. The sign of the coefficient of proportionality $\delta$ is consistent with Eq (5). In other words, as long as selection on unobservables is slightly larger (i.e. 16 %) than selection on observables it is enough for the true treatment effect to have the size of the IV estimates. This set could even seem conservative since officers are hired yearly per region. Hence, it is reasonable to expect that they might fluctuate relatively less compared to most crime variables.

## Concluding remarks

Considerable differences between the size of OLS and IV coefficients are likely seen as evidence against the validity of the instrument. The intuition supporting this viewpoint is that the OLS regression might be informative about the true effect the researcher wants to estimate. However, to our knowledge, in the literature there is no formal methodology to compare these two estimates.

To the best of our knowledge this is one of the first papers to suggest an objective criterion to compare IV and OLS estimates. For this purpose, this article adapts [1] setting to an IV framework. Furthermore, the analysis presented in this study disentangles in which setting this methodology might be used and suggests a way to implement it. Finally, this manuscript presents a simulated example and two observational examples to evaluate empirically its implementation.

This paper suggests that low values of the coefficient of proportionality offer supportive evidence that IV estimates are not *too large* with respect to OLS. The main limitation of the methodology presented in this paper is that high values of such a coefficient might either cast doubts on instrument validity or merely highlight that effects are heterogeneous [14,15]. To tackle this issue, this study suggests to complement the usage of this methodology with different analyses depending on the specifical setting.

The analysis carried out in this paper indicates that further research on criteria to objectively compare IV and OLS estimates is needed. In summary, this paper establishes a foundation for an objective criterion to compare IV and OLS estimates.

## Supporting information

**S1 Appendix.** See Sects A–C and Table A.1.
(PDF)

## Acknowledgments

I am deeply grateful to Juan José Dolado and five referees for their insightful comments on the results of this paper. Any remaining errors are, of course, my sole responsibility.

## Author contributions

**Conceptualization:** Riccardo Ciacci.

**Data curation:** Riccardo Ciacci.

**Formal analysis:** Riccardo Ciacci.

**Funding acquisition:** Riccardo Ciacci.

**Investigation:** Riccardo Ciacci.

**Methodology:** Riccardo Ciacci.

**Project administration:** Riccardo Ciacci.

**Resources:** Riccardo Ciacci.

**Software:** Riccardo Ciacci.

**Supervision:** Riccardo Ciacci.

**Validation:** Riccardo Ciacci.

**Visualization:** Riccardo Ciacci.

**Writing – original draft:** Riccardo Ciacci.

**Writing – review & editing:** Riccardo Ciacci.

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
