## [Decision Letter · Decision Letter 0]

28 Aug 2024

PONE-D-24-23558A Matter of Size: Comparing IV and OLS estimatesPLOS ONE

Dear Dr. Ciacci,

Thank you for submitting your manuscript to PLOS ONE. After careful consideration, we feel that it has merit but does not fully meet PLOS ONE’s publication criteria as it currently stands. Therefore, we invite you to submit a revised version of the manuscript that addresses the points raised during the review process.

We look forward to receiving your revised manuscript.

Kind regards,

Muhammad Amin

Academic Editor

PLOS ONE

Additional Editor Comments:

As the Reviewer 2 has serious concerns, therefore this require some major revisions. So, I recommend a major revision and incorporate all major/minor comments of the both reviewers.

Reviewers' comments:

Reviewer's Responses to Questions

**Comments to the Author**

1. Is the manuscript technically sound, and do the data support the conclusions?

Reviewer #1: Yes

Reviewer #2: Partly

2. Has the statistical analysis been performed appropriately and rigorously? 

Reviewer #1: Yes

Reviewer #2: No

3. Have the authors made all data underlying the findings in their manuscript fully available?

Reviewer #1: Yes

Reviewer #2: Yes

4. Is the manuscript presented in an intelligible fashion and written in standard English?

Reviewer #1: Yes

Reviewer #2: No

5. Review Comments to the Author

Reviewer #1: The submitted work addresses the discrepancies between Ordinary Least Squares (OLS) and Instrumental Variables (IV) estimates which are often seen as a red flag, suggesting potential issues with the validity of the instrument. Traditionally, these differences have been evaluated by directly comparing the OLS and IV coefficients, with little consideration given to the statistical information underlying the OLS estimates. This article introduces an approach by utilizing Oster's (2019) methodology to provide a more rigorous and objective framework for comparing OLS and IV estimates. A few examples, along with an extensive simulation study, validate the proposal. Therefore, this article holds a significant merit and should be considered for publication.

However, there is a minor observation: why did the author choose a panel data model for the presented framework?

Reviewer #2: Review report on PONE-2024-23558

“A Matter of Size: Comparing IV and OLS estimates”

In this paper, the author compares the significant differences between Ordinary Least Squares (OLS) and Instrumental Variables (IV) estimates, particularly in situations where the instrument(s) may be invalid. The framework of Oster’s (2019) methodology is also employed to contrast OLS and IV estimates. However, this article requires substantial revisions, as detailed below in the major and minor concerns.

Major Comments:

• The paper currently resembles a short report rather than a comprehensive academic study. It lacks sufficient detail on the topic of interest and the underlying motivation. For example, the primary objective of the paper should be clearly explained and linked to previous work. The author should clarify the research problem, identify the limitations of both the IV and OLS approaches, and outline what previous studies have accomplished in this area. Furthermore, the author’s contribution relative to existing studies should be explicitly stated in the introduction.

• In Section 2, just after Equation (4), the statement “Recall that if the instrument is valid, the IV coefficient consistently estimates β1” - the author should theoretically define the test used to determine the validity of instruments and report the type I and II error rates in the simulation results. Additionally, the concepts of “valid” and “invalid” instruments should be clearly defined, along with the potential consequences of using invalid instruments.

• The theoretical motivation for comparing OLS and IV estimates could be strengthened. The author should discuss the asymptotic properties of OLS and IV within the framework provided by Oster (2019).

• While the paper reports the point estimation of δ, it is unclear how the author tests the significance of δ. This aspect needs to be addressed.

• The paper should explore the behavior of the IV/OLS ratio when the instruments are weak. It would be interesting to include results for scenarios involving both strong and weak instruments in the simulation. Additionally, since real-world experiments often involve small sample sizes, I suggest that the author conduct simulations with smaller samples, starting with a sample size of 1000. Another simulation experiment could be added to explore different model specifications, including linear, nonlinear, and semiparametric models. Also, report the bias and RMSE.

Minor Comments:

• Although OLS and IV are common abbreviations, it would be helpful to spell them out in the abstract.

• The objective of the paper is not clearly articulated. If the first paragraph is intended to provide the motivation—stating that "Nonetheless, in the literature, to the best of my knowledge, there is no formal methodology to compare these two estimates"—then I must point out that the validity of instruments has been widely discussed in the literature. Many studies have compared the performance of OLS and Two-Stage Least Squares (TSLS), such as Murray (2006), Kolesár et al. (2015), and DiTraglia (2016), among others.

• The differences between Equation (1) and Equation (2) are not clear, as both seem to involve the same dependent variable yih. It appears that these models are intended to illustrate omitted variable bias. The author should clarify each model for the reader's benefit. There are different notations for the parameters (α1, α2, β1, β2, etc.), but not for the treatment variable and observed controls, which should be consistent.

• The author should clarify whether Model (1) is a reduced form model. Additionally, the first-stage regression and the instruments used in the model should be explicitly defined.

• In Section 3.1, the data-generating process for the simulation experiment should be clearly described. The author should specify the distribution of the treatment effect.

• Sections 4 and 5 should be moved to an online supplementary materials section, as they may not be essential to the main body of the paper.

Oster, E. (2019). Unobservable selection and coefficient stability: Theory and evidence.

Journal of Business & Economic Statistics 37(2), 187–204.

Murray, M. P. (2006). Avoiding invalid instruments and coping with weak instruments. Journal of economic Perspectives, 20(4), 111-132.

Kolesár, M., Chetty, R., Friedman, J., Glaeser, E., & Imbens, G. W. (2015). Identification and inference with many invalid instruments. Journal of Business & Economic Statistics, 33(4), 474-484.

DiTraglia, F. J. (2016). Using invalid instruments on purpose: Focused moment selection and averaging for GMM. Journal of Econometrics, 195(2), 187-208.

6. PLOS authors have the option to publish the peer review history of their article (what does this mean?). If published, this will include your full peer review and any attached files.

Reviewer #1: **Yes: **Muhammad Aslam

Reviewer #2: **Yes: **Muhammad Qasim

---

## [Author Response · Author response to Decision Letter 1]

27 Sep 2024

Dear Editor,

Please find attached a new version of the paper with the changes suggested by the referee reports in red. I also attach a response to each reviewer. As I explain in the corresponding file to Reviewer 2, I think there has been a misunderstanding. I have edited the paper following the suggestions yet the bulk of such comments are related to weak and invalid instruments. The paper does not deal with those because they fall outside the aim of the study. The article, indeed, provides a statistical methodology and framework to compare OLS and IV estimates once the researchers consider that the instrument is valid. Put it differently, if we are in an IV setting where scholars consider that the instrument is valid we can use the methodology developed in the paper to compare the size of OLS and IV estimates. Previously such comparison was made gauging the two by sight, after this paper this comparison might be carried out with a proper statistical framework. Yet, if the instruments are invalid, or specifically weak, this methodology cannot be used. As far as I understand, Reviewer 2 thought that my paper dealt with weak and invalid instruments and the comments made are written in that direction.

Thanks for your time, best regards

---

## [Decision Letter · Decision Letter 1]

10 Dec 2024

PONE-D-24-23558R1

A Matter of Size: Comparing IV and OLS estimates

PLOS ONE

Dear Dr. Ciacci,

Thank you for submitting your manuscript to PLOS ONE. After careful consideration, we have decided that your manuscript does not meet our criteria for publication and must therefore be rejected.

Specifically:

I am sorry that we cannot be more positive on this occasion, but hope that you appreciate the reasons for this decision.

Kind regards,

Muhammad Amin

Academic Editor

PLOS ONE

Additional Editor Comments:

In views of the serious concerns of the reviewer 2 and reviewer 3, this work is not acceptable for publication in Plos One

Reviewers' comments:

Reviewer's Responses to Questions

**Comments to the Author**

1. If the authors have adequately addressed your comments raised in a previous round of review and you feel that this manuscript is now acceptable for publication, you may indicate that here to bypass the “Comments to the Author” section, enter your conflict of interest statement in the “Confidential to Editor” section, and submit your "Accept" recommendation.

Reviewer #1: All comments have been addressed

Reviewer #3: All comments have been addressed

2. Is the manuscript technically sound, and do the data support the conclusions?

Reviewer #1: Yes

Reviewer #3: No

3. Has the statistical analysis been performed appropriately and rigorously? 

Reviewer #1: Yes

Reviewer #3: No

4. Have the authors made all data underlying the findings in their manuscript fully available?

Reviewer #1: Yes

Reviewer #3: Yes

5. Is the manuscript presented in an intelligible fashion and written in standard English?

Reviewer #1: Yes

Reviewer #3: No

6. Review Comments to the Author

Reviewer #1: As a minor observation in the earlier draft, it was asked that why did the author choose a panel data model for the presented framework. This observation has been addressed and the article is fit for acceptance.

Reviewer #3: This is a revision of the revised version of the paper which addresses the comments by the reviewers to the first version. I beg to agree with most comments by reviewer #2 and it is my view that the revised version did not address them in a satisfactory manner. My recommendation is not to publish the paper on the following grounds:

There is no doubt that the comparison of OLS and IV estimates is an important issue when someone may be discussing the robustness of estimates of treatment effects.

The purpose of the article is not clearly defined and the findings in the literature should be acknowledged, in particular Oster (2019). This paper is an implementation of the ideas in Oster (2019) and as such does not add anything new to current knowledge.

The paper provides examples for the implementation of the second way robustness statements might be made (i.e. making statements about R_max, (Oster, 2018, p. 196-197). I couldn’t locate the contributions the author claims to have added to the text. If the paper is an application of Oster’s ideas, at least the links with Imbens (2003) and Altonji, Elder & Taber (2005) should be described.

The issue of evaluating the robustness of results is intrinsically empirical. It is assumed that when designing a study, the researchers make decisions about what is an appropriate size for the treatment effect, how to control for confounders and which covariates should be included. Both theory and experience play an important role at this stage. Some of the confounders and covariates might not be observable at the time the study is being designed, and early decisions are made on how to deal with this problem (i.e. instrumental variables or any other approach). This brings me to the topic of the paper, which is how to assess the magnitude of biases due to unobserved variables and likely measurement errors on the estimation of treatment effects. A well-designed research would determine the size of the proportionality coefficient and the validity of candidates instruments in advance. Common sense suggests that the contribution of instruments to selection should be of a lower magnitude than that from the controls. This leave us with a desired value of 1 for the proportionality coefficient.

Most evaluations in the literature use quasi-experimental designs or are designed as observational studies. They use instruments as part of an identification strategy to introduce enough variability in the estimation of the treatment effects. Decisions on what a valid set of instruments should be, form part of the design process.

The paper does not include information on the context in which the Oster´s method is being applied. The point that the validity of instruments is a crucial assumption is something that needed to be stressed early in the introduction.

Later in the methodological section, there is continuous reference to the fact that the proportionality coefficient may indicate that the instruments might not be valid, a matter that causes confusion when reading the paper.

Before submission, the paper should have been reviewed by someone whose first language is English. There are lots of language problems.

7. PLOS authors have the option to publish the peer review history of their article (what does this mean?). If published, this will include your full peer review and any attached files.

Reviewer #1: **Yes: **Muhammad Aslam

Reviewer #3: No

- - - - -

---

## [Author Response · Author response to Decision Letter 2]

27 Jan 2025

As I exposed above there are two referees involved. One understood the paper and provided some comments to improve it. The other, unfortunately, did not understand the setting and suggested to reject my paper. By reading the referee report of this latter referee their mistakes are clear. In fact, the referee writes sentences of the sort of "the author should theoretically define the test used to determine the validity of instruments".

Furthermore, the referee misses the point of the article at all. The referee believes my article should deal with weak instruments (in the report the referee writes "The paper should explore the behavior of the IV/OLS ratio when the instruments are weak. It would be interesting to include results for scenarios involving both strong and weak instruments in the simulation."). Yet, my analysis cannot be adapted to weak instruments since the aim of the article is simply to provide a framework to compare the size of the IV and OLS coefficients. If the instrument is weak, then the instrument is invalid and the IV results might be misleading. Hence, one would need to rely on many more assumptions to compare these results with the OLS ones. Put succinctly, reading the reports of such referee it emerges that the referee did not understand the IV methodology and so cannot judge the article.

Thanks for your time and help, best regards.

---

## [Decision Letter · Decision Letter 2]

21 Mar 2025

PONE-D-24-23558R2A Matter of Size: Comparing IV and OLS estimatesPLOS ONE

Dear Dr. Ciacci,

Thank you for submitting your manuscript to PLOS ONE. After careful consideration, we feel that it has merit but does not fully meet PLOS ONE’s publication criteria as it currently stands. Therefore, we invite you to submit a revised version of the manuscript that addresses the points raised during the review process.

We look forward to receiving your revised manuscript.

Kind regards,

Muhammad Amin

Academic Editor

PLOS ONE

Journal Requirements:

Additional Editor Comments (if provided):

It recommended to incorporate all the reviewers comments.

Reviewers' comments:

Reviewer's Responses to Questions

**Comments to the Author**

1. If the authors have adequately addressed your comments raised in a previous round of review and you feel that this manuscript is now acceptable for publication, you may indicate that here to bypass the “Comments to the Author” section, enter your conflict of interest statement in the “Confidential to Editor” section, and submit your "Accept" recommendation.

Reviewer #2: (No Response)

Reviewer #3: (No Response)

2. Is the manuscript technically sound, and do the data support the conclusions?

Reviewer #2: Partly

Reviewer #3: Yes

3. Has the statistical analysis been performed appropriately and rigorously? 

Reviewer #2: N/A

Reviewer #3: Yes

4. Have the authors made all data underlying the findings in their manuscript fully available?

Reviewer #2: Yes

Reviewer #3: Yes

5. Is the manuscript presented in an intelligible fashion and written in standard English?

Reviewer #2: Yes

Reviewer #3: Yes

6. Review Comments to the Author

Reviewer #2: This is the revised paper submitted by the author(s). Based on the previous referee comments and the author(s)’ responses, it is evident that the paper has not been adequately revised according to the referee’s suggestions. Some of the referee’s comments are highly important and valid and should be addressed in the paper.

For example, the following two comments from the referee

1. "The paper should explore the behavior of the IV/OLS ratio when the instruments are weak. It would be interesting to include results for scenarios involving both strong and weak instruments in the simulation."

2. "Additionally, since real-world experiments often involve small sample sizes, I suggest that the author(s) conduct simulations with smaller samples, starting with a sample size of 1000. Another simulation experiment could be added to explore different model specifications, including linear, nonlinear, and semiparametric models. Also, report the bias and RMSE."

Some comments have only been partially addressed.

My observations ---

This study has several limitations, and the proposed methodology is not ideal for practitioners. For example, some reasonable comments were ignored by the author(s). Regardless of sample size, it is always good practice to report the bias and RMSE of the estimates.

One major limitation of this study is its heavy dependence on large datasets. The referee specifically requested an analysis with a sample size of 1000, but the author(s) have chosen not to include this. This omission suggests that if a researcher has a sample size of 1000 in their simulation, this methodology would not be suitable. The applicability of the proposed method is very narrow, which is a significant weakness of this paper.

Minor Comments

1. Table 7 note: The phrase "Notes: This table shows the bounds of the identified set using data from [?]."—what does "[?]" mean? Please clarify.

2. The first line lacks the source of the data in Example 2. Please provide a reference.

3. The structure of the model Model framework is not explicitly explained. Specifically, the dimensions of the variables and parameters in relation to their observations are unclear. For example, in "w is the vector of unobserved controls," what is the exact dimension of this vector? What is the dimension of the unobserved controls?

4. Equation (3): What do Var(X_ih) and Var(w_ih) represent? What is the formula for these variances, and how can the reader derive them to better understand the concept?

Reviewer #3: A Matter of Size: Comparing IV and OLS estimates

Manuscript Number: PONE-D-24-23558_R2_reviewer

Reviewer Comments

This is a second revision to the revised version of the paper which addresses the comments by the reviewers to the first version. I must clarify that my comments to the previous submission (Manuscript Number: PONE-D-24-23558) were entirely based on the comments by reviewer #2, therefore my recommendation was not publishing it. I must say that rather disqualifying my comments for being “empty” or showing lack of “understanding of the IV methodology” the author could have taken them as useful suggestions to improve the paper. In what follows, I will expand on each of the comments I provided in my previous review aiming the author would take them seriously. I think that the research reported in the paper contributes to the literature on comparing IV and OLS estimates, under the assumption that the instruments are valid. My current comments refer to my previous ones but relate to the Revised Manuscript with Track Changes attached to the files received from PLOS ONE.

Since the changes I suggest are easy to implement, my recommendation is to accept the paper for publication subject to making the changes suggested below.

There is no doubt that the comparison of OLS and IV estimates is an important issue when someone may be discussing the robustness of estimates of treatment effects.

The author qualified this comment as “empty” for not saying anything about the paper.

Perhaps, he might like to consider rewriting para 4 of the introduction along the following lines: “Comparison of OLS and IV estimates is an important issue when discussing the robustness of estimates of treatment or intervention effects when researchers use observables to assess the bias generated by unobservable variables in OLS settings (Murphy and Topel 1990; ….; De Luca et. Al. 2019). The main contribution of this paper is to provide a formal methodology to make such a comparison by extending that developed in Oster (2019). Previous research has addressed the issue but lacks methodological support (Alesina et. al. 2013; ...;Liu 2020).”

I strongly suggest to eliminate the reference to a “third contribution” as it contradicts the key assumption of instrument validity. It may confuse readers and distract them from the main purpose of the paper.

I suggest to stress the point that the proposed method is not aimed at assessing the validity of the instruments.

The purpose of the article is not clearly defined and the findings in the literature should be acknowledged, in particular Oster (2019). This paper is an implementation of the ideas in Oster (2019) and as such does not add anything new to current knowledge.

The version of the Revised Manuscript with Track Changes addressed this issue, at least partially.

The author might like to consider a more detailed elaboration on the extension to the case of comparing OLS and IV estimates. Oster´s method deals with the issue of unobserved controls but instrumental variables act like unobserved controls by allowing researchers to isolate the causal effect of a variable of interest.

The paper provides examples for the implementation of the second way robustness statements might be made (i.e. making statements about R_max, (Oster, 2018, p. 196-197). I couldn’t locate the contributions the author claims to have added to the text. If the paper is an application of Oster’s ideas, at least the links with Imbens (2003) and Altonji, Elder & Taber (2005) should be described.

Altonji, Elder & Taber (2005) (AET) was added to the list of references but not Imbens (2003) (American Economic Review,93: 126-132).

The author might like to mention somewhere in the introduction how Oster´s method relates to Imbens (2003) deals with the closely related logic behind partial least squares and AET proposed a method for evaluation of robustness under the assumption that the relationship between treatment and unobservables can be recovered from the relationship between treatment and observables.

The issue of evaluating the robustness of results is intrinsically empirical. It is assumed that when designing a study, the researchers make decisions about what is an appropriate size for the treatment effect, how to control for confounders and which covariates should be included. Both theory and experience play an important role at this stage. Some of the confounders and covariates might not be observable at the time the study is being designed, and early decisions are made on how to deal with this problem (i.e. instrumental variables or any other approach). This brings me to the topic of the paper, which is how to assess the magnitude of biases due to unobserved variables and likely measurement errors on the estimation of treatment effects. A well-designed research would determine the size of the proportionality coefficient and the validity of candidates instruments in advance. Common sense suggests that the contribution of instruments to selection should be of a lower magnitude than that from the controls. This leave us with a desired value of 1 for the proportionality coefficient.

I regret the author did qualify this comment as “empty”. At least he might like to take some of the ideas above to provide some background to the response that the aim of this paper is to remove “common sense”.

Most evaluations in the literature use quasi-experimental designs or are designed as observational studies. They use instruments as part of an identification strategy to introduce enough variability in the estimation of the treatment effects. Decisions on what a valid set of instruments should be, form part of the design process.

Again, the author considered this as an “empty” comment. The proposed method rests on the assumption of instrument validity and the paper shows how to implement it with panel data (Observational data ,Examples 1 and 2).

I suggest the author to include a text similar to that in the comment as background for the referred section. Keeping in mind that PLOS ONE is not an econometrics-economics journal and that the potential readers might not be familiar with modern micro econometrics approaches, a sort of an introduction to this section will provide a reason for the examples beyond the “complexity” in the data.

The paper does not include information on the context in which the Oster´s method is being applied. The point that the validity of instruments is a crucial assumption is something that needed to be stressed early in the introduction.

The author replied by arguing about my lack of knowledge of IV techniques.

Both of us know that instruments may not be valid and that an IV analysis often only estimates the “local treatment effect” which may not generalize to the entire population.

Despite the author´s disqualification of this reviewer econometrics credentials, the issue addressed by this comment was stated at the end of the first paragraph of the introduction.

What would happen when this is not the case?

Later in the methodological section, there is continuous reference to the fact that the proportionality coefficient may indicate that the instruments might not be valid, a matter that causes confusion when reading the paper.

This issue was addressed by reviewer #2 and I believe that it continues to be a weakness of the paper.

Since your method provides a point value for delta and has not distributional properties associated to it, how can you assess whether its value is large enough to indicate either the invalidity of the instrument, or heterogenous effects, or both?

Before submission, the paper should have been reviewed by someone whose first language is English. There are lots of language problems.

My comment was based on PLOS ONE criterion 1 for publication: “PLOS ONE does not copyedit accepted manuscripts, so the language in submitted articles must be clear, correct, and unambiguous. We may reject papers that do not meet these standards. If the language of a paper is difficult to understand or includes many errors, we may recommend that authors seek independent editorial help before submitting a revision. These services can be found on the web using search terms like “scientific editing service” or “manuscript editing service.””

It is not a bad idea to check the quality of the writing with an expert.

7. PLOS authors have the option to publish the peer review history of their article (what does this mean?). If published, this will include your full peer review and any attached files.

Reviewer #2: No

Reviewer #3: No

---

## [Decision Letter · Decision Letter 3]

20 Aug 2025

PONE-D-24-23558R3A Matter of Size: Comparing IV and OLS estimatesPLOS ONE

Dear Dr. Ciacci,

Thank you for submitting your manuscript to PLOS ONE. After careful consideration, we feel that it has merit but does not fully meet PLOS ONE’s publication criteria as it currently stands. Therefore, we invite you to submit a revised version of the manuscript that addresses the points raised during the review process.

As Reviewer 4 recommended a substantial revision, I recommend a major revision. After incorporating all the reviewers' major and minor concerns, the work can be accepted for publication. 

We look forward to receiving your revised manuscript.

Kind regards,

Muhammad Amin

Academic Editor

PLOS ONE

Journal Requirements:

Additional Editor Comments (if provided):

As Reviewer 4 recommended a substantial revision, I recommend a major revision. After incorporating all the reviewers' major and minor concerns, the work can be accepted for publication.

Reviewers' comments:

Reviewer's Responses to Questions

**Comments to the Author**

1. If the authors have adequately addressed your comments raised in a previous round of review and you feel that this manuscript is now acceptable for publication, you may indicate that here to bypass the “Comments to the Author” section, enter your conflict of interest statement in the “Confidential to Editor” section, and submit your "Accept" recommendation.

Reviewer #3: All comments have been addressed

Reviewer #4: All comments have been addressed

Reviewer #5: (No Response)

2. Is the manuscript technically sound, and do the data support the conclusions?

Reviewer #3: Yes

Reviewer #4: Partly

Reviewer #5: Yes

3. Has the statistical analysis been performed appropriately and rigorously? 

Reviewer #3: Yes

Reviewer #4: Yes

Reviewer #5: Yes

4. Have the authors made all data underlying the findings in their manuscript fully available?

Reviewer #3: Yes

Reviewer #4: No

Reviewer #5: Yes

5. Is the manuscript presented in an intelligible fashion and written in standard English?

Reviewer #3: Yes

Reviewer #4: (No Response)

Reviewer #5: Yes

6. Review Comments to the Author

Reviewer #3: A Matter of Size: Comparing IV and OLS estimates

Manuscript Number: PONE-D-24-23558_R3_reviewer

Reviewer Comments

The author has addressed all my comments in a satisfactory manner. With the changes made to the previous version, I believe the paper is in a format that is acceptable for publication..

There is no doubt that the comparison of OLS and IV estimates is an important issue when someone may be discussing the robustness of estimates of treatment effects.

Addressed.

The purpose of the article is not clearly defined and the findings in the literature should be acknowledged, in particular Oster (2019). This paper is an implementation of the ideas in Oster (2019) and as such does not add anything new to current knowledge.

Addressed.

The paper provides examples for the implementation of the second way robustness statements might be made (i.e. making statements about R_max, (Oster, 2018, p. 196-197). I couldn’t locate the contributions the author claims to have added to the text. If the paper is an application of Oster’s ideas, at least the links with Imbens (2003) and Altonji, Elder & Taber (2005) should be described.

Addressed.

The issue of evaluating the robustness of results is intrinsically empirical. It is assumed that when designing a study, the researchers make decisions about what is an appropriate size for the treatment effect, how to control for confounders and which covariates should be included. Both theory and experience play an important role at this stage. Some of the confounders and covariates might not be observable at the time the study is being designed, and early decisions are made on how to deal with this problem (i.e. instrumental variables or any other approach). This brings me to the topic of the paper, which is how to assess the magnitude of biases due to unobserved variables and likely measurement errors on the estimation of treatment effects. A well-designed research would determine the size of the proportionality coefficient and the validity of candidates instruments in advance. Common sense suggests that the contribution of instruments to selection should be of a lower magnitude than that from the controls. This leave us with a desired value of 1 for the proportionality coefficient.

Addressed.

Most evaluations in the literature use quasi-experimental designs or are designed as observational studies. They use instruments as part of an identification strategy to introduce enough variability in the estimation of the treatment effects. Decisions on what a valid set of instruments should be, form part of the design process.

Addressed

The paper does not include information on the context in which the Oster´s method is being applied. The point that the validity of instruments is a crucial assumption is something that needed to be stressed early in the introduction.

Addressed

Later in the methodological section, there is continuous reference to the fact that the proportionality coefficient may indicate that the instruments might not be valid, a matter that causes confusion when reading the paper.

Addressed

Before submission, the paper should have been reviewed by someone whose first language is English. There are lots of language problems.

Addressed

Reviewer #4: Title: A Matter of Size: Comparing IV and OLS estimates

Manuscript ID: PONE-D-24-23558R3

This is a technically solid and methodologically focused paper that proposes an objective framework—based on Oster (2019)—to compare IV and OLS estimates. The motivation is clear: large differences between IV and OLS coefficients are often interpreted as evidence of instrument invalidity, but without a formal benchmark. By formalizing this comparison and illustrating it through both simulations and observational applications, the author fills a methodological gap in applied econometrics.

There are some issues that the authors need to address before the manuscript can be considered for publication. The following are my comments describing these issues.

1- Contribution and Relevance

• The contribution is methodological and addresses a gap in applied econometrics practice: a formal, reproducible approach for OLS–IV comparison.

• The integration with Oster’s method is original in this context, and the emphasis on implementability (including Stata code) is a major strength.

• The simulations convincingly show that large IV/OLS ratios do not necessarily imply instrument invalidity.

2- Theoretical Framework

• The derivation from equations (1)–(5) is correct but dense. Some intermediate economic intuition would help applied readers, e.g., explaining in words:

o Why the proportionality assumption is plausible in some settings.

o What kinds of omitted variables w are captured by δ in applied terms.

• The paper moves quickly from the univariate to the multivariate setting; a table summarizing symbols, assumptions, and their roles would improve accessibility.

3- Assumption Discussion

• The proportional selection assumption (equation (3)) is strong. While the author mentions that prior knowledge of the setting is crucial, more discussion is needed on:

o Situations where the assumption is likely violated (e.g., when observables are weak proxies for unobservables).

o Sensitivity of results to mis-specification of Rmax.

o Possible robustness checks (e.g., alternative Rmax values, bounding δ using prior studies).

4- Simulation Study

• The Monte Carlo section is informative, but the practical meaning of thresholds such as ∣δ∣<1 or ∣δ∣<2 is not fully explained.

The reader would benefit from a short “practical guidelines” subsection translating these findings into applied advice (e.g., what value of ∣δ∣ should start to raise concern).

5- Empirical Applications

• Acemoglu et al. (2001):

o The conclusion that no plausible δ can reconcile OLS and IV estimates is striking. The discussion could more explicitly connect this to historical debates on settler mortality and institutions to help contextualize why IV estimates may be inflated.

• Ciacci (2024):

o This example benefits from a clear rationale for the instrument (flights), but robustness to alternative specifications or placebo outcomes would strengthen the case.

o Since this is the author’s own prior work, a note on possible self-selection in variable choice could be made transparent.

6- Interpretation and Limitations

• The paper is careful to state that large δ values could indicate either invalid instruments or heterogeneous effects. However:

o A more structured decision tree could help applied researchers interpret results (e.g., “If δ is large, check X, Y, Z…”).

o The policy implications of such findings could be briefly discussed — this would broaden the readership beyond methodologists.

Reviewer #5: Comments on A Matter of Size: Comparing IV and OLS estimates

This paper analyses a framework where Oster (2019)’s methodology

might be used to compare OLS and IV estimates. This methodology of-

fers evidence to support or discard IV estimates with respect to the OLS

regression.

1. There is something fundamentally wrong with the sentence ”A com-

mon approach in econometrics is to interpret substantial differences

between the magnitudes of Ordinary Least Squares (OLS) and Instru-

mental Variables (IV) coefficients as evidence against the validity of the

instrument.” If we have a problem where we need to use IV this implies

that OLS is inconsistent and we should expect differences between IV

and OLS.

2. ”IV is an econometric methodology primarily used to estimate a causal

effect” This is not true. IV have many uses in many various fields and

some are in the causal framework but I shouldn’t claim that it is the

main use.

3. I think the paper would benefit from being more clear that we have

the following situation. There are situations where OLS is inconsistent

while IV is not. We went to know if this is the case and we develop a

strategy to do so. Further, as we know if OLS is consistent and Iv is

consistent then the OLS is more efficient so we should use that. This

implies that we can use a Hausman-Wu type of test.

4. There are many instances where equation should be Equation

5. Equation 4 and 5. It seems very strange that there is a hat on β2.

Thins that you should rewrite this in terms of population parameters

for OLS and for IV respectively, i.e. do not speak about estimates. You

have population covariances etc to do this! this would also solve the

problem with ”true effect β1, the IV estimates” as estimates are not

true...

6. Thins you really need a good discussion about the ”proportional selec-

tion assumption” and if this assumption is reasonable.

7. Simulated data example. Need to present exactly how data is gener-

ated.

8. Table 1: It is very strange that standard deviation is the same while

the mean differs.

9. The research topic of this paper is related to bounds on selection bias,

see e.g. Smith, LH, VanderWeele, TJ. Bounding bias due to selec-

tion. Epidemiology 2019;30:509–16 and Zetterstrom, S, Waernbaum, I.

Selection bias and multiple inclusion criteria in observational studies.

Epidemiol Methods 2022;11:1–21. I think the Authors can expand the

introdution to include a more diverse discussion.

10. The observational data examples are good as informative examples. I

would like that the Author adds a discussion why the IV estimates are

outside the reasonable bounds, i.e. what can we learn from this exercise

besides the facts that there is an inconsistency between the OLS and

the IV estimates.

11. There is one point I think that the Author misses and that is that

even though the IV estimates are not valid, that does not imply that

the OLS estimates are fine,i.e. the are only informative in that respect

that they can help identify if the IV estimator is good enough.

12. After reading the whole paper I think I now understand what the Au-

thor want to state in the introduction but I think the fails doing so. I

think he can be more clear that the research problem here is that the

IV estimates sometimes are much further away from the OLS estimates

than is warranted by e.g. omitted variable bias.

7. PLOS authors have the option to publish the peer review history of their article (what does this mean?). If published, this will include your full peer review and any attached files.

Reviewer #3: No

Reviewer #4: **Yes: **Hounaida Daly

Reviewer #5: No

---

## [Editor Report · Decision Letter 4]

29 Sep 2025

A Matter of Size: Comparing IV and OLS estimates

PONE-D-24-23558R4

Dear Dr. Ciacci,

We’re pleased to inform you that your manuscript has been judged scientifically suitable for publication and will be formally accepted for publication once it meets all outstanding technical requirements.

Kind regards,

Muhammad Amin

Academic Editor

PLOS ONE

Additional Editor Comments (optional):

Based on reviewers' reports, this work can be considered for publication after incorporating the minor concerns as suggested Hounaida Daly.
---

## [Editor Report · Acceptance letter]

PONE-D-24-23558R4

PLOS ONE

Dear Dr. Ciacci,

I'm pleased to inform you that your manuscript has been deemed suitable for publication in PLOS ONE. Congratulations! Your manuscript is now being handed over to our production team.

Kind regards,

on behalf of

Dr. Muhammad Amin

Academic Editor

PLOS ONE